# Nanostructure and Optical Property Investigations of SrTiO_3_ Films Deposited by Magnetron Sputtering

**DOI:** 10.3390/ma12010138

**Published:** 2019-01-03

**Authors:** Da Xu, Yafei Yuan, Huanfeng Zhu, Ling Cheng, Chunmin Liu, Jing Su, Xintong Zhang, Hao Zhang, Xia Zhang, Jing Li

**Affiliations:** 1Department of Optical Science and Engineering, Shanghai Engineering Research Center of Ultra-Precision Optical Manufacturing, Fudan University, Shanghai 200433, China; 09210720010@fudan.edu.cn (D.X.); 15110720031@fudan.edu.cn (Y.Y.); 10110720035@fudan.edu.cn (H.Z.); 16210720009@fudan.edu.cn (L.C.); 18110720032@fudan.edu.cn (C.L.); 16110720001@fudan.edu.cn (J.S.); 17110720006@fudan.edu.cn (X.Z.); zhangh@fudan.edu.cn (H.Z.); 2Center for Intelligent Medical Electronic Engineering, School of Information Science and Technology, Fudan University, Shanghai 200433, China; 3Key Laboratory of Micro and Nano Photonic Structures (Ministry of Education), Fudan University, Shanghai 200433, China; 4College of physical Engineering, Qufu Normal University, Qufu 273165, China; xzhangqf@126.com

**Keywords:** strontium titanate film, optical properties, thermal treatment, magnetron sputtering

## Abstract

Strontium titanate thin films were deposited on a silicon substrate by radio-frequency magnetron sputtering. The structural and optical properties of these films were characterized by X-ray diffraction, high-resolution transmission electron microscopy, X-ray photoelectron spectroscopy, and spectroscopic ellipsometry, respectively. After annealing at 600–800 °C, the as-deposited films changed from amorphous to polycrystalline. It was found that an amorphous interfacial layer appeared between the SrTiO_3_ layer and Si substrate in each as-deposited film, which grew thicker after annealing. The optical parameters of the SrTiO_3_ film samples were acquired from ellipsometry spectra by fitting with a Lorentz oscillator model. Moreover, we found that the band gap energy of the samples diminished after thermal treatment.

## 1. Introduction

Strontium titanate (SrTiO_3_) has attracted much interest in many fields for its excellent properties, such as high dielectric constant, low leakage current, good dielectric tenability, and high Seebeck coefficient. As one of the perovskite oxides, SrTiO_3_ has a face-centered cubic structure at room temperature. In the bulk of SrTiO_3_, its lattice constant is 3.90 Å, and the indirect band gap is 3.2 eV, which is expected to originate from the separation between the 2*p*-level of oxygen ions and the 3*d*-level of titanium ions [1]. As an excellent dielectric film for Dynamic Random Access Memories (DRAMs), SrTiO_3_ thin film has a high dielectric constant, even when very thin (~10 nm) [2]. A Resistive Random Access Memory (RRAM) cell fabricated using SrTiO_3_ thin film has a large resistance ratio, up to 10^3^–10^4^ between high and low resistance states, and shows good retention properties in a long test time [3]. Regarding dielectric tunable devices, SrTiO_3_ thin film has a 65% variation of the permittivity in the terahertz range [4]. Because of its large Seebeck coefficient, SrTiO_3_ thin film is also an attractive thermoelectric material [5,6].

SrTiO_3_ thin films can be fabricated by many depositional techniques. Chemical techniques include atomic layer deposition (ALD) [7], metal organic vapor deposition (MOCVD) [8], and the sol-gel process [9], among others. Physical techniques include pulsed laser deposition (PLD) [10] and magnetron sputtering [11,12,13], among others. Compared to other techniques, magnetron sputtering shows many advantages, i.e., wide compositional versatility, very high purity, extremely high adhesion of films, controllable deposition rate, etc. Although previous studies have reported on the optical and electrical properties of different SrTiO_3_ thin films, few investigations have mentioned the influence of thermal treatment on the microstructural and optical properties. In this study, SrTiO_3_ thin films deposited by radio-frequency (RF) magnetron sputtering were annealed at different temperatures (500–800 °C), and the influence of thermal treatment on the crystallization, surface morphology, cross-section structure, film chemistry, and optical properties were investigated in detail.

## 2. Materials and Methods

The SrTiO_3_ thin films were deposited on silicon substrates with <100> single crystalline orientation at room temperature using a LAB600sp typed RF magnetron sputtering system (Leybold Optics GmbH, Dresden, Germany). The size of SrTiO_3_ target with 99.99% purity was 4 inches in diameter and 6 mm in thickness. The background pressure in the vacuum chamber was 5.0 × 10^−6^ mbar. The RF power was set to 75 W. The working pressure of Ar gas was 9.6 × 10^−3^ mbar controlled by a mass flowmeter (MFC, Bronkhorst High-Tech B.V., Ruurlo, The Netherlands). After deposition, four samples were annealed at different temperatures of 500 °C, 600 °C, 700 °C and 800 °C in nitrogen for one hour, respectively.

The crystallinity of the as-deposited SrTiO_3_ thin films annealed at 500–800 °C was characterized by X-ray diffraction (XRD, Rigaku, Neu-Isenburg, Germany) with a Rigaku D/MAX 2550 VB/PC typed X-ray diffractometer using Cu *K*_α_ radiation (*λ* = 1.5406 Å). The surface roughness of the films was measured using a PSIA XE-100 atomic force microscope (AFM, PSIA, Suwon, Korea). High-resolution transmission microscopy (HRTEM, FEI, Hillsboro, OR, USA) is capable of imaging at a significantly higher resolution to capture fine detail, even as small as a single column of atoms, owing to the smaller de Broglie wavelength of electrons. The specimen is most often an ultrathin section less than 100 nm thick or a suspension on a grid. HRTEM was employed to examine the cross-sectional microstructure of the film samples. The depth profiles and the chemical binding structures of the sample films were studied using X-ray photoelectron spectroscopy (XPS, Thermo Fisher Scientific, Waltham, MA, USA) with Al K_α_ X-rays and *E*_photon_ = 1500 eV. XPS can be used to analyze the surface chemistry of a material in its as-received state or after some treatment, such as thermal treatment. The optical constants, including refractive index and extinction coefficient, of the films were determined by spectroscopic ellipsometry (SE, Self-development, Shanghai, China) in the spectral range from 280 nm to 800 nm. Moreover, the band gap energy was calculated from the SE spectra.

## 3. Results and Discussions

Figure 1 shows the XRD patterns of as-deposited and annealed SrTiO_3_ thin films at different temperatures. No characteristic peaks of the SrTiO_3_ layer can be found in both the as-deposited and 500 °C annealed samples. It indicates both of the films are amorphous. When the annealing temperature reaches 600 °C, the thin film became polycrystalline, which is proved by the emergence of the three characteristic peaks at (100), (110), and (200). It shows that the onset crystallization temperature is around 600 °C. As the annealing temperature increases to 700 °C and 800 °C, the diffraction peaks became more intense and sharper, showing enhanced crystallinity of the SrTiO_3_ samples. Meanwhile, the lattice constant can be calculated from the diffraction peaks, which is *a* = *b* = *c* = 3.91 Å, showing cubic structure. Moreover, the average grain size can be determined from the major peak (200) by Scherrer’s formula [14],
(1)D=κλ/Bcosθ
where *D* is the average grain size, *λ* is the X-ray wavelength, *B* is the full width at half maximum of the peak, *θ* is the diffraction angle, and *κ* is the Scherrer’s constant of the order of unity for usual crystals. The average grain sizes of the thin films were 16.9 nm, 21.2 nm and 26.7 nm, corresponding to annealing temperatures of 600 °C, 700 °C and 800 °C, respectively.

Figure 2 shows the cross-section HRTEM micrographs of SrTiO_3_ samples which are, respectively, as-deposited and annealed at temperature from 500 to 800 °C. As shown in Figure 2a, the as-deposited SrTiO_3_ thin film is amorphous. The thickness of the SrTiO_3_ layer is 75.54 nm, as shown in Table 1. Meanwhile, an interfacial layer between the SrTiO_3_ layer and Si substrate is also observed, and the thickness of this layer measured in Figure 2b is 3.76 nm. After annealing at 500 °C, SrTiO_3_ thin film is still amorphous, and the thicknesses of SrTiO_3_ layer and interfacial layer are 74.69 nm and 2.53 nm, respectively. After annealing at 600 °C, crystallization took place in the SrTiO_3_ layer, as can be seen in Figure 2f. The inhomogeneity of SrTiO_3_ thin film decreases and the surface roughness increases, which is in good agreement with the XRD results. The thickness of the interfacial layer increases to 3.08 nm. One reason for this increase is the penetration of the particles from SrTiO_3_ layer and Si substrate into the interfacial layer as an effect of annealing, as confirmed below. When the annealing temperature is increased to 700 °C and 800 °C, the inhomogeneity of SrTiO_3_ thin films decreases sequentially, and the thicknesses of the interfacial layers increase to 5.58 nm and 12.22 nm, respectively.

To further investigate the elemental composition and chemical states of the SrTiO_3_ and interfacial layers in all the samples, XPS analysis was carried out. Figure 3 is the concentration depth profiles of the as-deposited and 800 °C annealed samples. The thickness of the SrTiO_3_ layer in as-deposited sample is thicker than that in 800 °C annealed sample. Besides, after annealing, more Si and SrTiO_3_ diffused to interfacial layer, causing the thickness of interfacial layer to increase. These results are in good agreement with the HRTEM results. For the as-deposited sample, the ratio of Sr/Ti/O in the etching time range of 0–840 s is about 1:1:3, indicating that the SrTiO_3_ thin film deposited by RF magnetron sputtering is reliable.

Figure 4a,b show Sr 3*d* core levels and Ti 2*p* core levels XPS spectra, respectively, of the as-deposited and 800 °C annealed SrTiO_3_ thin films at the etching time of 360 s. In Figure 4a, the peaks at 133.9 and 135.5 eV correspond to the binding energies of Sr^2+^ 3*d*_5/2_ and 3*d*_3/2_ [12]. In Figure 4b, the peaks at 458.8 and 464.3 eV correspond to the binding energies of Ti^4+^ 2*p*_3/2_ and 2*p*_1/2_ [12,15]. Another two weak peaks at 457.3 and 463.1 eV correspond to the binding energies of Ti^3+^ 2*p*_3/2_ and 2*p*_1/2_ [15], which come from defects in SrTiO_3_ thin films. These defects were reduced after annealing at 800 °C.

Figure 5 shows Si 2*p* core levels XPS spectra of the as-deposited and 800 °C annealed SrTiO_3_ thin films at the etching time of 960 s. The Si 2*p* peak is centered at around 99.2 eV, and the SiO_2_ peak is located at about 103.0 eV, which are consistent with previous results [16,17]. After annealing at 800 °C, the intensity of Si 2*p* peak is found to decrease greatly, while the intensity of SiO_2_ peak increases. This change in intensities is caused by the diffusion and reaction of silicon and oxygen in the SrTiO_3_/Si interface.

To investigate the optical constants and band-gap structure of SrTiO_3_ thin films prepared at different temperatures, the spectroscopic ellipsometry (SE) technique is applied in the range of 290–800 nm with different angles of incidence at 65°, 70°, and 75° [18]. The ellipsometric parameters 𝛹 and Δ are defined as,
(2)$$\rho =r_{s}/r_{p}=tan\Psi exp\left(i\Delta \right)$$
where *r_p_* and *r_s_* represent the complex reflection coefficients of polarized light parallel and perpendicular to the incidence plane, respectively. Since the roughness layer of the as-deposited and 500 °C annealed samples are very thin (<1.8 nm), a four-phase model of Si substrate/interfacial layer/SrTiO_3_ layer/Air is designed for these two samples, and a five-phase model of Si substrate/interfacial layer/SrTiO_3_ layer/roughness layer/Air is designed for the samples annealed at 600 °C, 700 °C, and 800 °C. The effective complex dielectric function ε of the roughness layer can be parameterized using the Maxwell-Garnett effective medium approximation (EMA) presented as,
(3)$$\frac{\varepsilon -\varepsilon_{Air}}{\varepsilon +2\varepsilon_{Air}}=\frac{\varepsilon_{STO}-\varepsilon_{Air}}{\varepsilon_{STO}+2\varepsilon_{Air}}f$$
where $\varepsilon_{Air}$ (~1) and $\varepsilon_{STO}$ are dielectric functions of atmosphere and SrTiO_3_ thin film, respectively, and *f* is the volume fraction of SrTiO_3_ in the roughness layer. Two Lorentz oscillators model and single Lorentz oscillator model are used to characterize $\varepsilon_{STO}$ and $\varepsilon_{IL}$ (dielectric function of the interfacial layer) [19], respectively, described as follows,
(4)$$\varepsilon \left(E\right)=\varepsilon_{1}+i\varepsilon_{2}=\varepsilon \left(\infty \right)\left(1+\underset{i}{\sum }\frac{A_{i}^{2}}{E_{i}^{2}-E^{2}-j\Gamma_{i}E}\right)$$
where ε(∞) is the dielectric constant when photon energy *E*→∞, *A_i_*, *Γ_i_*, and *E_i_* are, respectively, the amplitude, the damping factor, and center energy of the *i*th oscillator in units of eV. The refractive index and extinction coefficient can be calculated from the dielectric function as follows,
(5)$$n={\left[\frac{1}{2}\sqrt{\varepsilon_{1}^{2}+\varepsilon_{2}^{2}}+\varepsilon_{1}\right]}^{\frac{1}{2}}$$
(6)$$k={\left[\frac{1}{2}\sqrt{\varepsilon_{1}^{2}+\varepsilon_{2}^{2}}-\varepsilon_{1}\right]}^{\frac{1}{2}}$$

In the fitting process, the thickness of each layer is fixed on the value in Table 1.

Figure 6 shows the calculated refractive indices and extinction coefficients of SrTiO_3_ thin films, both as-deposited and at different annealing temperatures. The parameters of the Lorentz oscillator model for SrTiO_3_ thin film are listed in Table 2. As can be seen in Figure 6, there are two dispersion regions in the range from 1.55 to 4.42 eV: one is the transparent region (1.55–4.00 eV for the as-deposited and 500 °C annealed SrTiO_3_ film, and 1.55–3.50 eV for SrTiO_3_ film annealed at 600–800 °C), and the rest region is the absorption region. Figure 6a shows that the refractive index increases with annealing temperature before annealing at 700 °C, and decreases with annealing temperature after annealing at 700 °C. Such a change is attributed to different values of packing density *p* of the films [9,10], which can be calculated from Lorentz-Lorenz relation [20],
(7)$$p=\left(\frac{n^{2}-1}{n^{2}+2}\right)/\left(\frac{n_{b}^{2}-1}{n_{b}^{2}+2}\right)$$
where *n_b_* is the refractive index of bulk SrTiO_3_. Taking *n_b_* = 2.432 at 550 nm, the values of packing densities are 0.86, 0.87, 0.96, 0.94 and 0.93 for the as-deposited film and films annealed at 500 °C, 600 °C, 700 °C and 800 °C, respectively. The as-deposited SrTiO_3_ film has a minimum packing density, which increases a little after annealing at 500 °C. The increase in packing density will lead to a decrease in thickness of SrTiO_3_ layer, as shown in Table 1. When the annealing temperature is up to 600 °C, the packing density increases to a maximum value, which is caused by the crystallization of SrTiO_3_ film. After annealing at 700 °C and 800 °C, the packing density shows a tendency of decreasing, which is attributed to the presence of cracks in the films at higher annealing temperatures.

As shown in Figure 6b, the extinction coefficients are very small (<0.02) in the transparent region. Meanwhile, the absorption edge moves toward lower photon energy at higher annealing temperatures. The absorption peak of the SrTiO_3_ thin film in high photon energy regions comes from the electronic inter-band transition [9]. Hence, the movements of absorption edges are related to the varieties of the bandgap structures in SrTiO_3_ thin films. The SrTiO_3_ thin film’s indirect-band-gap [21] and band gap *E_g_* can be determined from the power-law behavior of Tauc [22],
(8)$${\left(\alpha h\nu \right)}^{1/2}=C\left(h\nu -E_{g}\right)$$
where *α* is the absorption coefficient, *hν* is the photon energy, and *C* is a constant. The absorption coefficient can be calculated from the relation,
(9)$$\alpha =\frac{4\pi k}{\lambda }$$

Figure 7 shows the dependence of ${\left(\alpha h\nu \right)}^{1/2}$ on hν for SrTiO_3_ thin films prepared at different temperatures. The band gap *E_g_* is then determined by extrapolating the linear portion of the curves in the limit ${\left(\alpha h\nu \right)}^{1/2}=0$. The values of the band gap *E_g_* are listed in Table 2. As evident from Figure 7 and Table 2, the band gap decreases as the annealing temperature increases. The as-deposited and 500 °C annealed SrTiO_3_ thin films have a similar amorphous structure, and thus, the difference between their band gaps is small. After annealing at 600 °C, the SrTiO_3_ thin film transits from an amorphous phase to a polycrystalline phase, which results in a large decrease in band gap. When the annealing temperature goes up to 700 °C and 800 °C, the band gap decreases due to better crystallinity of SrTiO_3_ thin film.

## 4. Conclusions

SrTiO_3_ thin films have been deposited on Si substrates using RF magnetron sputtering, and then annealed in air at temperatures from 500 °C to 800 °C for 1 h. The as-deposited SrTiO_3_ thin film is amorphous with a stoichiometric ratio of about 1:1:3. The RMSE roughness of the as-deposited film is only 0.23 nm. The transition from amorphous phase to polycrystalline phase occurred at an annealing temperature between 500 °C and 600 °C. With the increase of annealing temperature, the average grain size and surface roughness of SrTiO_3_ thin films increase, while the inhomogeneity decreases. The refractive index in the transparent region increases with annealing temperature until 700 °C, and then decreases. The band gaps are estimated to be about 4.11, 4.08, 3.80, 3.75, and 3.71 eV for the as-deposited and annealed at 500 °C, 600 °C, 700 °C, and 800 °C SrTiO_3_ thin films, respectively. These results are useful as references for the potential applications of SrTiO_3_ in integrated optical and electrical devices.

## Figures and Tables

**Figure 1 materials-12-00138-f001:**
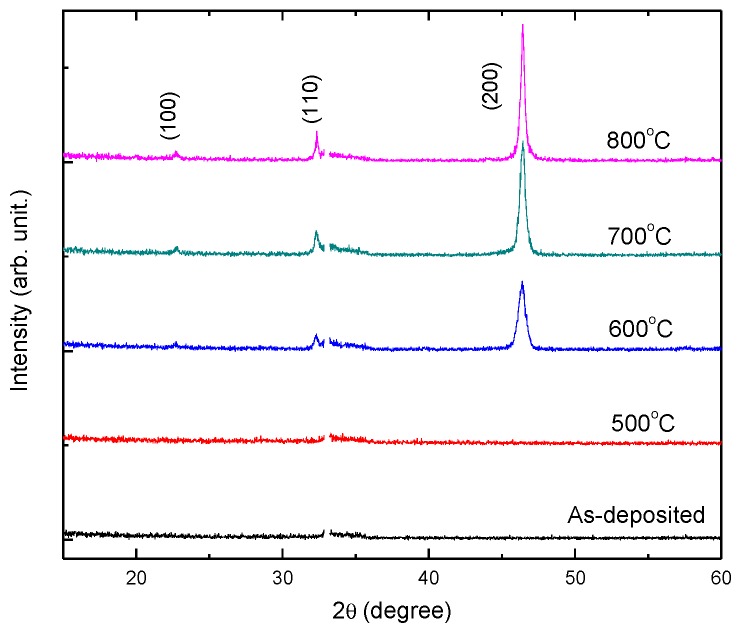
X-ray diffraction (XRD) patterns of SrTiO_3_ thin film annealed at different temperatures.

**Figure 2 materials-12-00138-f002:**
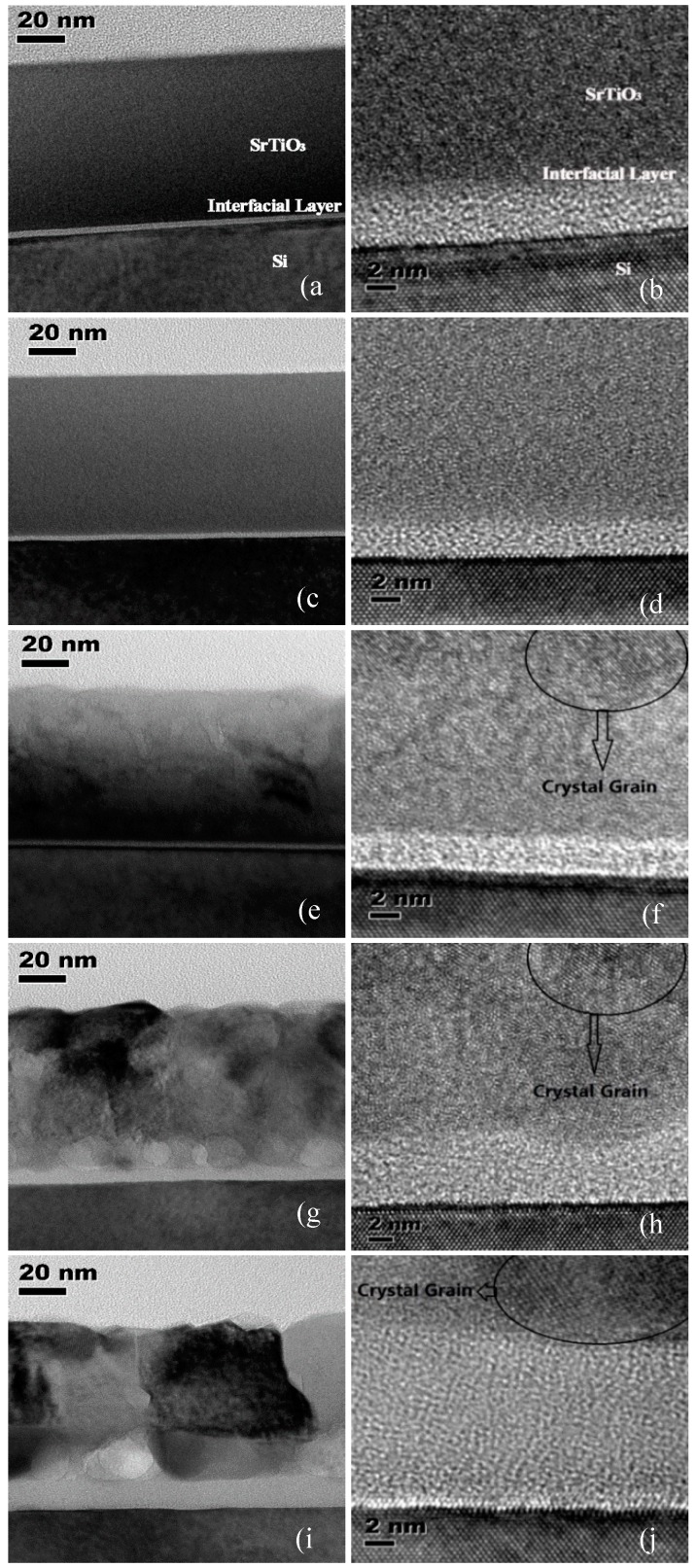
High-resolution transmission microscopy (HRTEM) photographs of SrTiO_3_ films (**a**) as-deposited, (**c**) 500 °C annealed, (**e**) 600 °C annealed, (**g**) 700 °C annealed, (**i**) 800 °C annealed, and the zoom in interfacial layers (**b**) as-deposited, (**d**) 500 °C annealed, (**f**) 600 °C annealed, (**h**) 700 °C annealed, (**j**) 800 °C annealed.

**Figure 3 materials-12-00138-f003:**
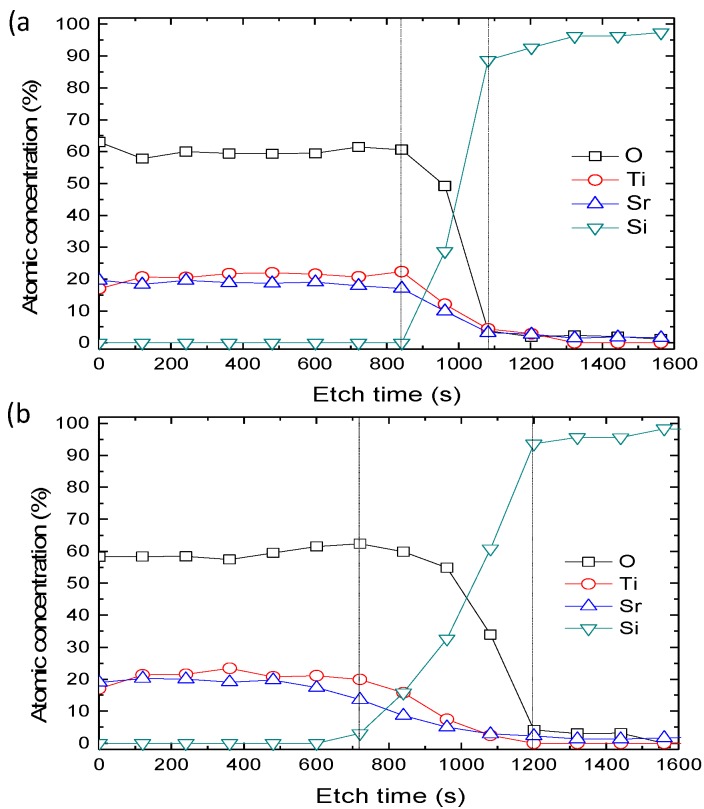
Depth profiles of the SrTiO_3_ films (**a**) as-deposited and (**b**) annealed at 800 °C.

**Figure 4 materials-12-00138-f004:**
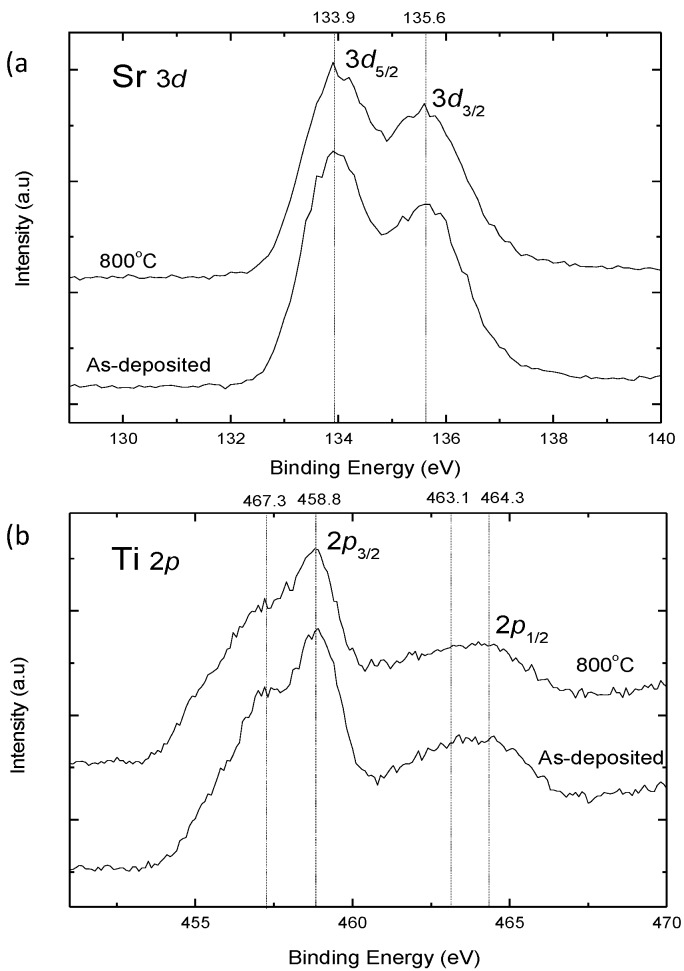
The X-ray photoelectron spectroscopy (XPS) spectra of (**a**) Sr 3*d* and (**b**) Ti 2*p* in the SrTiO_3_ films as-deposited and annealed at 800 °C.

**Figure 5 materials-12-00138-f005:**
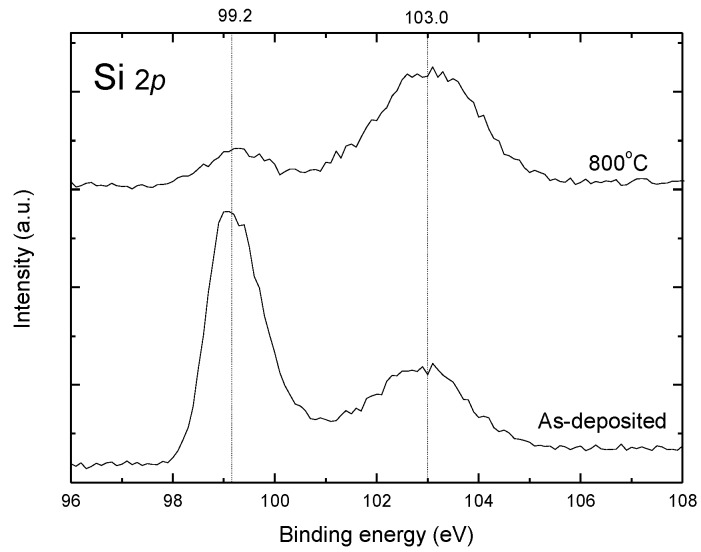
The Si 2*p* core levels XPS spectra of the as-deposited and 800 °C annealed SrTiO_3_ thin films at the etching time of 960 s.

**Figure 6 materials-12-00138-f006:**
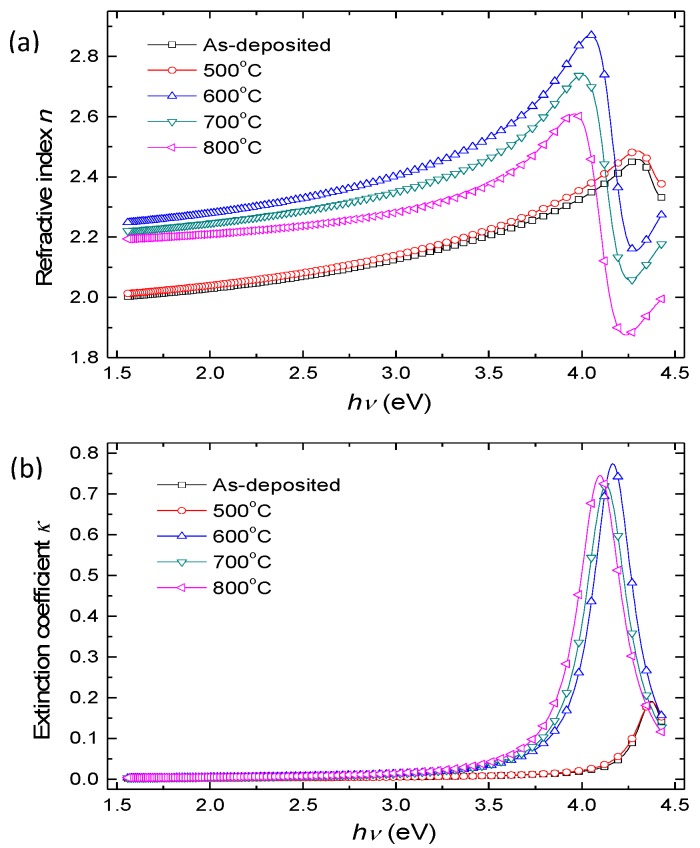
The calculated (**a**) refractive indices and (**b**) extinction coefficients of SrTiO_3_ thin films prepared at different temperatures.

**Figure 7 materials-12-00138-f007:**
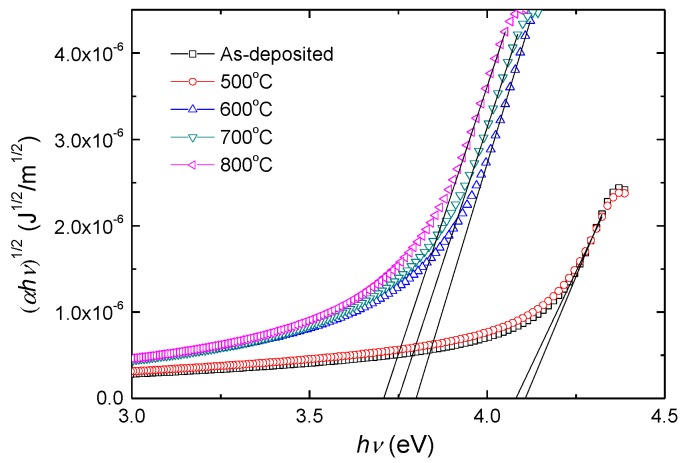
The ${\left(\alpha h\nu \right)}^{1/2}$ vs. hν for the SrTiO_3_ films annealed at different temperatures.

**Table 1 materials-12-00138-t001:** Thickness data of the SrTiO_3_ films annealed at different temperatures.

Sample	SrTiO_3_ Layer (nm)	Interfacial Layer (nm)	Roughness Layer (nm)	Root-Mean-Square Error (RMSE) Roughness (nm)
As-deposited	75.54	3.76	1.41	0.23
500 °C	72.96	2.53	1.73	0.33
600 °C	67.09	3.08	6.91	1.58
700 °C	68.21	5.58	7.77	1.76
800 °C	67.60	12.22	8.31	1.94

**Table 2 materials-12-00138-t002:** Main parameters (ε(∞), $A_{i}$ and $E_{i}$) of the Lorentz oscillator model and the calculated band gap $E_{g}$ for SrTiO_3_ thin films annealed at different temperatures.

Sample	ε(∞)	$A_{1}\;(\mathrm{eV})$	$E_{1}\;(\mathrm{eV})$	$A_{2}\;(\mathrm{eV})$	*E*_2_ (eV)	*E_g_* (eV)
As-deposited	1.51	0.67	4.37	8.11	6.63	4.11
500 °C	1.63	0.65	4.37	7.47	6.45	4.08
600 °C	1.50	1.39	4.15	14.02	9.20	3.80
700 °C	1.68	1.51	4.11	10.19	7.80	3.75
800 °C	1.73	1.48	4.08	13.70	10.80	3.71

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
