# Peer review of "Nanostructure and Optical Property Investigations of SrTiO_3_ Films Deposited by Magnetron Sputtering"

_materials, 2019, doi:10.3390/ma12010138_

Round 1

Reviewer 1 Report

The paper presents new results for deposition of SrTiO3 films. There are some shortcomings which should be removed prior to publication.

Line 33: is it a direct or an indirect band gap?

Line 53-56: what kind of Si substrate was used (crystallinity, orientation, purity, …)? What kind of target (composition, size, thickness, purity) was employed? How did you set the working gas pressure?

Line 55: delete “controlled at”.

Line 64: some details regarding the XPS equipment are required.

Line 109: that the film ….. is stoichiometric?

Figure 4: the spectrum looks fairly strange and should be discussed in context with previous works, e.g., M.C. Biesinger, L.W.M. Lau, A. Gerson and R.St.C. Smart, Resolving Surface Chemical States in XPS Analysis of First Row Transition Metals, Oxides and Hydroxides: Sc, Ti, V, Cu and Zn, Applied Surface Science, 257 (2010) 887-898, see http://www.xpsfitting.com/2008/09/titanium.html

Figures 3-5: a peak fitting to extract individual contributions should be applied.

Line 137: what does “pretty” mean in this context?

Line 143: dielectric functions “in” air?

Line 161: how is it possible that the transparent region extends to 4 eV (3 eV?) when the extinction coefficient at 4 eV is largest?

Line 165/167: how is the packing density defined? Where does a packing density less than unity come from?

Table 2: delete insignificant digits (considering the errors). 2 or 3 digits should be more reasonable.

Author Response

Please find the attachment as the point-by-piont reponse to your comments. Thanks for your help.

Reviewer 2 Report

The article contains new and significant data. The material is of scientific interest.

There are some mistakes. Please correct.

1)divide Chapter 2 into two: 1 Experimental Part & 2 Results & Discussions

2) transfer all the details of the methodology to the "Experimental Part" (all formulas 1-9 and their descriptions, etc.)

3) Add descriptions of XPS and HRTEM study

4) Use in the text abbriviatures of "XPS" (not "XP")

5) Add SAED to the Fig. 2. as a additional column.

6) Add 3D images of surface (AFM data) as additional column on Fig. 2

7) Detalised the roughness data in separate paragraph (I see inf. about Ra just in Conclusions)

8)  Table 2. Information in digits isn't visible. Exlude the overlapping of digits.

Author Response

(The authors gave the same response as above.)

Round 2

Reviewer 1 Report

The authors have considered all points. The paper is now ready for publication.